Neglected Tropical Diseases

# The mosaic of Rickettisae from the Caatinga Biome, Brazil: New records and first interactions

João F. Audi Gazeta[1]*, Ana Beatriz Pais Borsoi[2], Karla Bitencourth[2], Mariana Guimarães Côrtes[2], Ingrid Benevides Machado[2], Beatriz Pinheiro Melo da Silva[1], Robson Cavalcante[3], Gilberto Salles Gazeta[2]*, Nathalie Costa da Cunha[2]

**1** Departamento de Saúde Coletiva Veterinária e Saúde Pública, Universidade Federal Fluminense, Niterói, Rio de Janeiro, Brazil, **2** Laboratório de Referência Nacional em Vetores das Riquetsioses, Instituto Oswaldo Cruz (IOC-FIOCRUZ), Rio de Janeiro, Brazil, **3** Laboratório de Vetores, Reservatórios e Animais Peçonhentos Dr. Thomaz Corrêa Aragão (LAVRAP), Fortaleza, Ceará, Brazil

* joaofgazeta@hotmail.com (JAG); gsgazeta@ioc.fiocruz.br (GSG)

## Abstract

Rickettsiosis are infections that have worldwide distribution, also occurring in all regions of Brazilian territory and being distributed across its various biomes. The most relevant rickettsiosis in Brazil is Spotted Fever (SF), a potentially lethal zoonosis. Although cases of SF have been confirmed in southeastern Brazil, the epidemiological situation in the Northeast remains poorly understood. The aim of this study was to analyze the circulation and identification of *Rickettsia* species in the state of Ceará, a northeastern state with confirmed SF cases since 2010. In parallel, we identified potential vectors, as to contribute to the development of an eco-epidemiological profile of Rickettsiae in the state. Genomic DNA samples, extracted from ticks collected in areas with suspected cases, were analyzed via PCR to detect *Rickettsia* spp., with positive samples undergoing further characterization and genomic sequencing for *ompB* (SFG-IF/SFG-TG IR) and *htrA* (17Kd1/17Kd2) genes via Nested PCR. Our results detected, for the first time, the presence of *Rickettsia amblyommatis* and *R. asembonensis* in the region. These findings demonstrated the geographical expansion of Rickettsiae in northeastern Brazil. Additionally, we identified the first interaction between *Rickettsia parkeri* and *Amblyomma parvum*, signaling a potential change in our understanding of interactions between these species. Altogether, our study reveals a new panorama regarding Rickettsiae in northeastern Brazil.

**Data availability statement:** The data genetic data that support the findings of this study will be publicly available from the National Center for Biotechnology Information (NCBI) library (GENBANK) for acession with the identifier(s) (PV085523-PV085565) upon the manuscript acceptance/publication.

**Funding:** The author(s) received no specific funding for this work.

**Competing interests:** The authors have declared that no competing interests exist.

## Author summary

Rickettsiae are bacteria spread mainly by ticks and fleas that can cause diseases in humans and animals. This study focused on identifying different species of Rickettsia and their potential tick carriers in the state of Ceará, Brazil. We examined 1,359 samples from ticks collected on animals and in the environment. After initial testing, 82 samples showed signs of Rickettsia bacteria, and we successfully identified specific species in 37 of them using genetic analysis. Notably, this research found *Rickettsia amblyommatis* and *Rickettsia asembonensis* circulating in the region for the first time, adding to our understanding of local bacteria and their tick hosts. We also discovered a new association between the tick *Amblyomma parvum* and *Rickettsia parkeri*, a species linked to mild human infections. These findings help map which ticks carry which bacteria, offering insights into how Rickettsiae spread in the environment. This is crucial for public health planning, as some species can cause illness in humans. By identifying these bacteria and their carriers, this research contributes to a better understanding of disease risks in Ceará and similar regions worldwide.

## Introduction

Rickettsioses are zoonotic infectious diseases caused by bacteria of the genus *Rickettsia*, pleomorphic organisms with an obligate intracellular life cycle, transmitted by wingless arthropods with global distribution [1]. Among the known forms of rickettsioses, the Spotted Fever Group (SFG) zoonoses, primarily represented in Brazil by the species *Rickettsia rickettsii* and *Rickettsia parkeri*, hold the greatest significance and distribution. These diseases are considered of mandatory notification in Brazil due to their acute nature and potentially lethal outcomes [2].

Species of the genus *Rickettsia* are currently classified within four groups: Ancestral Group, Transitional Group, Typhus Group, and Spotted Fever Group (SFG). Except for the Ancestral Group, all others include pathogenic species capable of causing diverse clinical conditions in humans and animals. Notably, *Rickettsia prowazekii* and *Rickettsia typhi* are the causative agents of epidemic typhus and murine typhus, respectively, while *R. rickettsii* and *R. parkeri* are the primary species linked to spotted fever in the Americas [1]. *Rickettsia* species from the Ancestral and Spotted Fever Groups are predominantly associated with ticks, whereas those from the Typhus and Transitional Groups are more commonly linked to mites, fleas, and lice [3,4].

In Brazil, the presence of vectors capable of maintaining and transmitting rickettsioses has been confirmed in all regions, along with the detection of pathogenic *Rickettsiae*, leading to the implementation of the **National Environmental Surveillance System for Rickettsioses and Other Tick-Borne Diseases** (Rede Nacional de Vigilância Ambiental para as Rickettsioses e outras Doenças Transmitidas por Carrapatos – RNVADTC). This initiative has enhanced the understanding of the country's epidemiological situation regarding this public health concern. The primary vectors of rickettsioses in Brazil are ticks, blood-feeding ectoparasites represented

by various genera and species, such as *Amblyomma spp.*, *Rhipicephalus spp.*, and *Dermacentor spp.*, all of which have been associated with *Rickettsia* spp. Other potential vectors of *Rickettsia* in Brazil include fleas, lice, and mites [5].

**Spotted fever in Brazil: an overview**

With the establishment of Spotted Fever (SF) as a notifiable disease and the implementation of the surveillance network, new studies on its true distribution and epidemiological status began to emerge. Initially, these studies were more focused on the South and Southeast regions of Brazil but eventually expanded to previously neglected areas, particularly the Midwest, North, and Northeast regions. In light of this increased surveillance, data gathered from the Notifiable Diseases Information System (SINAN/Ministry of Health) reveals that the Southeast region has experienced the highest number of confirmed cases in the historical series (2007–2024), totaling 1,924 cases (approximately 113 cases per year). The South region follows with 726 confirmed cases (approximately 43 cases per year), while the Northeast reports 56 cases, the Midwest has 38, and the North, the least affected, recorded 10 cases during this period [6].

In this context, the most severe form of spotted fever, associated with *R. rickettsii* infection, is recorded in the Southeast region and the northern part of Paraná, a state in the South region of the country [7] In these areas, predominantly composed of Cerrado (Tropical Savanna) and anthropized Atlantic Rainforest biomes, the transmission of *R. rickettsii* is mainly attributed to the tick *Amblyomma sculptum*, especially in environments with the presence of equids and capybaras, the main hosts of this vector [8]. Additionally, a study conducted by Pacheco et al. [9] in the state of Minas Gerais identified *Rhipicephalus linnaei* ticks (previously classified as *R. sanguineus*) naturally infected with *R. rickettsii* and capable of transmission, with the etiological agent being isolated and cultured in the laboratory.

In contrast, a focus of Spotted Fever was identified in an area with lower anthropogenic impact in the metropolitan region of São Paulo. In this scenario, the tick *Amblyomma aureolatum* stands out as the primary vector of *R. rickettsii*, while dogs appear to play a significant role in the epidemic cycle of the disease. This distinct dynamic reflects the ecological and behavioral differences that can influence the transmission of Spotted Fever in various regions of the country [8,10].

Although the epidemiological landscape in certain regions of Brazil is relatively well characterized, the understanding of Spotted Fever (SF) in other environments remains incipient. For instance, in the Midwest region of Brazil, there have been reports of cases with notable clinical differences, but without confirmation of the vector species and rickettsiae involved. Likewise, in other areas of the country identified as new hotspots for SF, such as Maranhão, Rondônia, Pernambuco, Alagoas, Paraíba, and Sergipe, further studies are needed to clarify the epidemiological profile of the region's rickettsiae [7,11].

In the Northeast region, in which stands the Caatinga biome, the eco-epidemiological landscape of spotted fever, as well as of rickettsioses in general, tough somewhat more familiar, remains relatively unclear. In 2010, the Ceará State Health Secretariat reported the first confirmed cases of spotted fever in the Baturité Massif and the capital city, Fortaleza— both of which are tropical climates and fall within the Atlantic Forest biome, in contrast to the majority of the state, which is characterized by a semi-arid climate and the Caatinga biome. In 2016, further studies confirmed the circulation of *Rickettsiae* from different groups in the state, as well as the presence of potential vectors [12]. Also, recent findings from Cavalcante et al. [13] now indicate the presence of rickettsiae in several municipalities within the Ceará semi-arid region. Therefore, this study aimed to analyze the epidemiological situation of rickettsioses in the state, with a particular focus on the biodiversity of the etiological agent involved and its main vectors.

## Methods

The work was carried out at the Molecular Epidemiology Laboratory, at the Veterinary School of the Federal Fluminense University (Laboratório de Epidemiologia Molecular, Faculdade de Veterinária - Universidade Federal Fluminense); the Tick and Other Apterous Arthropods Laboratory – National Reference in Rickettsial Vectors (Laboratório de Carrapatos e Outros Artrópodes Ápteros, LAC – Fiocruz); and the Fiocruz Technological Platforms for Genomic Sequencing Network (PDTIS).

## Vector collection and transport

All samples used in this study were obtained from the LAC sample repository, through the National Environmental Surveillance System for Rickettsioses and Other Tick-Borne Diseases (RNVADTC). The capture of potential vector samples was carried out during environmental surveillance and case investigations for Spotted Fever between 2017 and 2023 by the Health Secretariat of the State of Ceará (SESA/CE) and LAC members. Potential vectors were gathered from vertebrate hosts and the environment using techniques such as flannel dragging and direct picking. All samples were preserved in analytical grade isopropanol (≥99% purity) immediately after collection and transported in UN3373-compliant containers for biological samples, following LAC established internal biosafety protocols for long-term DNA preservation and taxonomic work.

## Morphological identification and storage

Captured ticks were then sent to the Central Laboratory of the State of Ceará (LACEN/CE) and subsequently forwarded to the LAC, following the protocol established by the RNVADTC. Taxonomic identification was performed under a stereomicroscope at the time of each sample's arrival at the LAC–FIOCRUZ laboratory, following each collection event between 2017 and 2023, using specific identification keys: Amorim and Serra-Freire [14] for the larval stage, Martins et al. [15] for the nymphal stage, and Barros-Battesti et al. [16] and Dantas-Torres et al. [17] for the adult stage. After identification and cataloging, tick samples were stored in analytical grade isopropanol at -20°C at the laboratory repository.

## DNA extraction, amplification and sequencing

Identified specimens were aliquoted into 1.5 mL microtubes, either individually (for nymphs and adults) or grouped (for larvae). Subsequently, the samples underwent genomic DNA (gDNA) extraction following the enzymatic extraction protocol previously described by Aljanabi and Martinez [18] in 1997. In accordance with LAC's protocol, only a portion of collected tick samples (30–50%) collected from each host were screened for *Rickettsia* genes, while the remaining specimens were preserved for potential confirmatory analysis (counter-proofing).

The DNA was then subjected to amplification through screening PCR (Polymerase Chain Reaction) using genus-specific and group-specific markers for SFG Rickettsiae: *gltA* (CS2–78/CS2–323) and *ompA* (Rr.190.70p/Rr190.602), respectively [19]. Samples that amplified at least one of these genes were further analyzed using specific characterization markers for Rickettsiae from the spotted fever and typhus groups: *htrA* (17Kd1/17Kd2) and *ompB* (SFG-IF/SFG-TG IR) [20,21]. After amplification, extracted DNA samples were stored in purified (Milli-Q) water at -80°C in the repository, in accordance with internal protocol.

The shift in primers was motivated by the conditions of older archived samples, many of which had undergone genetic degradation over time due to prolonged storage. As a result, amplification and sequencing using the initial *gltA* and *ompA* primers was often unsuccessful, particularly in older samples. To address this, we employed nested PCR protocols targeting the aforementioned *htrA* and *ompB* fragments in all samples, which offered improved sensitivity and better species-level differentiation. This approach also allowed us to standardize the molecular workflow, particularly through the use of *ompB*, which provides greater discriminatory power among closely related spotted fever group rickettsiae during sequencing.

After the amplification of each gene, the samples were subjected to electrophoresis on a 2% agarose gel, stained with ethidium bromide [22]. The gel visualization and photographic documentation were performed using the MiniBis Pro system (DNR Bio - Imaging Systems Ltd). Samples that amplified the expected fragment were purified using the Wizard SV Gel and PCR Clean-up System (Promega) kit, following the manufacturer's recommendations. Subsequently, the purified amplicons were sent for sequencing of the obtained genetic material.

DNA sequencing was performed on the DNA sequencing platform (PDTIS) at FIOCRUZ/RJ. Sequencing reactions were carried out using the BigDye Terminator v3.1 Cycle Sequencing Kit (Applied Biosystems, Carlsbad, California, USA), following the manufacturer's recommendations. The same primers used in the PCR were employed in these reactions for

sequence determination in both directions (3'-5' and 5'-3'). Subsequently, the samples were precipitated, resuspended in formamide, and applied to an ABI 3730xl automatic sequencer (Applied Biosystems, Carlsbad, California, USA) for sequence reading.

## Molecular data analysis

All obtained sequences for each gene were manually edited, and consensus sequences were generated using the ChromasPro 1.5 program (Technelysium Pty Ltd, Tewantin, Qld, Australia). Initially, the DNA sequences were identified by similarity assessment, through a comparative analysis with sequences deposited in GenBank, using BLASTN (Basic Local Alignment Search Tool - Nucleotide).

After editing, the sequences obtained were automatically aligned using the ClustalW multiple alignment algorithm [23], available in the MEGA 6.0 program [24]. All alignments were manually inspected, and the protein-coding genes were translated into amino acids to check for the presence of pseudogenes and confirm homology, with no stop codons being observed.

To improve upon the understanding of the identification of rickettsiae detected in the Ixodidae ticks studied in Ceará, a concatenated phylogenetic reconstruction of fragments from the *ompB* (411 bp) and *htrA* (434 bp) genes was performed. The SeaView program [25] was used to concatenate the alignments of the investigated genes. Phylogenetic analysis, using maximum likelihood analysis, was carried out in the PhyML 3.0 program [26] with the evolutionary model T92 + G (through the Bayesian Information Criterion) indicated by MEGA 6.0, which best fitted the analyzed dataset. Statistical support values for internal branches were estimated with the aLRT test (approximate likelihood ratio test) using 1000 replicates [27].

## Results

During the study period (2017–2023), environmental surveillance activities and investigations of suspected Spotted Fever cases were conducted by the Health Secretariat of the State of Ceará (SESA/CE), as well as municipal and regional health secretariats across 14 municipalities in the state. The distribution of infested hosts recorded during these activities is presented in Table 1 below.

A total of 3,098 ticks with vector potential were collected during the study period, comprising 1,562 females (50.4%), 1,125 males (36.3%), 378 nymphs (12.2%), and 33 larvae (1%). Of this total, 1,362 (44%) were processed according to the protocol established by LAC - FIOCRUZ, of which 82 (6%) amplified for *Rickettsia* sp. fragments during preliminary

**Table 1.  Absolute (n) and relative (%) frequency of the main vertebrate hosts found with potential vectors for *Rickettsia* sp. during previous surveillance activities in different municipalities of the state of Ceará, from 2017 to 2023.**

| Vertebrate Host | Host commom name | n | % |
|---|---|---|---|
| *Canis familiaris* | Domestic dog | 338 | 75,11% |
| *Bos taurus* | Cattle | 63 | 14% |
| *Homo sapiens* | Human | 19 | 4,22% |
| *Felis catus* | Domestic Cat | 9 | 2% |
| *Equus caballus* | Horse | 9 | 2% |
| *Gallus galuus* | Chicken | 6 | 1,33% |
| *Cerdocyon thous* | Crab-eating Fox | 3 | 0,67% |
| *Coendou* sp. | Porcupine | 3 | 0,67% |
| **Total** | | **450** | **100%** |

screening for *gltA* and/or *ompA.* The distribution of species in relation to hosts, along with their respective absolute and relative frequencies, can be found in Table 2 (Ixodidae/Argasidae) below

Among the 1,362 samples processed during the study period, 82 (6%) tested positive during screening, showing amplification for the target genes *gltA* and/or *ompA*. From the 82 samples that amplified during screening, genetic sequencing and phylogenetic analysis were successfully performed on 37, representing 2.72% of the total and 45.12% of those that amplified the expected fragment. Regarding additional genetic targets, 25 samples amplified for the *ompB* gene through Nested PCR, of which 24 were successfully sequenced and analyzed. Similarly, 23 samples amplified for the *htrA* gene, with 21 yielding successful sequencing results. Notably, 8 samples were successfully sequenced for both *ompB* and *htrA* gene fragments.

Following sequencing and phylogenetic analysis, a concatenated phylogenetic tree was constructed using the aforementioned fragments (Fig 1). This analysis revealed the presence of five circulating *Rickettsia* species in the studied regions: *Candidatus Rickettsia andeanae* (n = 16), *Rickettsia amblyommatis* (n = 9), *Rickettsia parkeri* strain Atlantic Rainforest (n = 7), *Rickettsia felis* (n = 4), and *Rickettsia asembonensis* (n = 1).

The *Rickettsia* species were detected in a variety of vectors (six species) originating from different hosts, as shown in Tables 3 and 4 (which include municipalities previously associated with Rickettisae) below. Notably, of the 37 samples that were successfully sequenced, 14 (47.82%) were found in domestic dogs, 7 (29.17%) were from vectors collected directly from the environment and not associated with specific hosts, 2 (8.69%) were obtained from armadillos (*Euphractus sexcinctus*), 2 (8.69%) from cattle (*Bos taurus*), 1 (4.34%) from a hedgehog (*Coendou prehensilis*), and 1 (4.34%) from a specimen of *Amblyomma parvum*, which was found parasitizing a human.

The collected samples were distributed from 14 municipalities across various biomes, microregions, and mesoregions of the state. The study revealed the presence of rickettsiae in three new municipalities: Tauá, Trairi, and Jijoca do Jericoacoara. Additionally, in the regions of Guaiuba, Santana do Cariri, Itapipoca, Tururu, and Tianguá, the research contributed to expanding the knowledge of circulating rickettsiae by identifying specific species in areas where rickettsiae were previously detected but not identified on a species level. In the remaining municipalities, the research contributed to expanding knowledge on the circulating rickettsiae, revealing the presence of new species in areas where rickettsiae had already been detected previously. The geographic distribution of Rickettsia species identified in tick samples during the study is presented in Fig 2 below.

## Discussion

Among the five *Rickettsia* species identified in this study, *R. felis* and *R. asembonensis* are classified within the Transitional Group, while *Rickettsia parkeri*, *Rickettsia amblyommatis*, and *Candidatus Rickettsia andeanae* are members of the Spotted Fever Group (SFG). The detection of *R. parkeri*, *R. felis*, and *Ca. R. andeanae* corroborates previous findings reported by Moerbeck et al. [12,28] and Cavalcante et al. [13]. Notably, there have been no prior records of *R. asembonensis* or *R. amblyommatis* in studies conducted within the state, thereby establishing this study as the first to document the presence of these organisms in the region.

### Spotted fever group rickettsiae

**Rickettsia parkeri.** *Rickettsia parkeri* was identified in three distinct genetic clusters: *Rickettsia parkeri sensu stricto* (s.s.), *R. parkeri* Atlantic Rainforest strain, and *R. parkeri* NOD strain. These clusters represent clades within the species, of which *R. parkeri s.s.* and *R. parkeri* Atlantic Rainforest strain have confirmed pathogenicity, causing a mild form of spotted fever. In contrast, the pathogenicity of the NOD strain remains unknown [29,30].

It is important to note that during the phylogenetic analysis, it was observed that there is no available sequence in GenBank for *Rickettsia parkeri* strain NOD for the analyzed *ompB* gene fragment, which limited the analysis of the results, since samples LIC 10044B, LIC 10044A, and LIC 11559C only successfully sequenced the *ompB* fragment.

**Table 2. Absolute (n) and relative (%) frequency of ticks (Ixodidae/Argasidae) captured during previous surveillance activities in different municipalities of the state of Ceará, from 2017 to 2023.**

| Host | Ixodidae | | | | | | | | Argasidae | | | Total |
|---|---|---|---|---|---|---|---|---|---|---|---|---|
| | A. longi-rostre | A. nodosum | A. ovale | A. parvum | A. rotun-datum | D. nitens | Rh. linnaei | Rh. microplus | Argas miniatus | O. rietcorreai | O. talaje | |
| | n (%) | n (%) | n (%) | n (%) | n (%) | n (%) | n (%) | n (%) | n (%) | n (%) | n (%) | n (%) |
| Environment | 6 (18,75) | - | 1 (3,22) | 15 (39,50) | 2 (50) | - | 185 (18,3) | - | 25 (80) | 8 (100) | - | 242 (17,77) |
| C. familiaris | - | 2 (100) | 26 (83,88) | 13 (34,20) | - | - | 816 (80,7) | 6 (3,05) | - | - | - | 863 (63,36) |
| Bos taurus | - | - | - | 2 (5,26) | - | - | 4 (0,40) | 189 (95,94) | - | - | - | 195 (14,32) |
| Homo sapiens | - | - | 3 (9,68) | 5 (13,15) | 2 (50) | - | - | 2 (1,01) | - | - | 3(100) | 15 (1,10) |
| Felis catus | - | - | - | 1 (2,63) | - | - | 4 (0,40) | - | - | - | - | 5 (0,37) |
| E. caballus | - | - | - | - | - | 5 (100) | - | - | - | - | - | 5 (0,37) |
| Gallus galuus | - | - | - | - | - | - | - | - | 6 (20) | - | - | 6 (0,44) |
| Ce. thous | - | - | 1 (3,22) | 2 (5,26) | - | - | 2 (0,20) | - | - | - | - | 5 (0,37) |
| Coendou sp. | 26 (81,25) | - | - | - | - | - | - | - | - | - | - | 26 (1,91) |
| Total | 32 (100%) | 2 (100%) | 31 (100%) | 38 (100%) | 4 (100%) | 5 (100%) | 1011 (100%) | 197 (100%) | 31(100%) | 8(100%) | 3 (100%) | 1362 (100%) |

A: *Amblyomma*; Rh: *Rhipicephalus*; D: *Dermacentor*; C: *Canis*; Ce: *Cerdocyon*; E: *Equus*; O: *Ornithodoros*; (-): Not found.

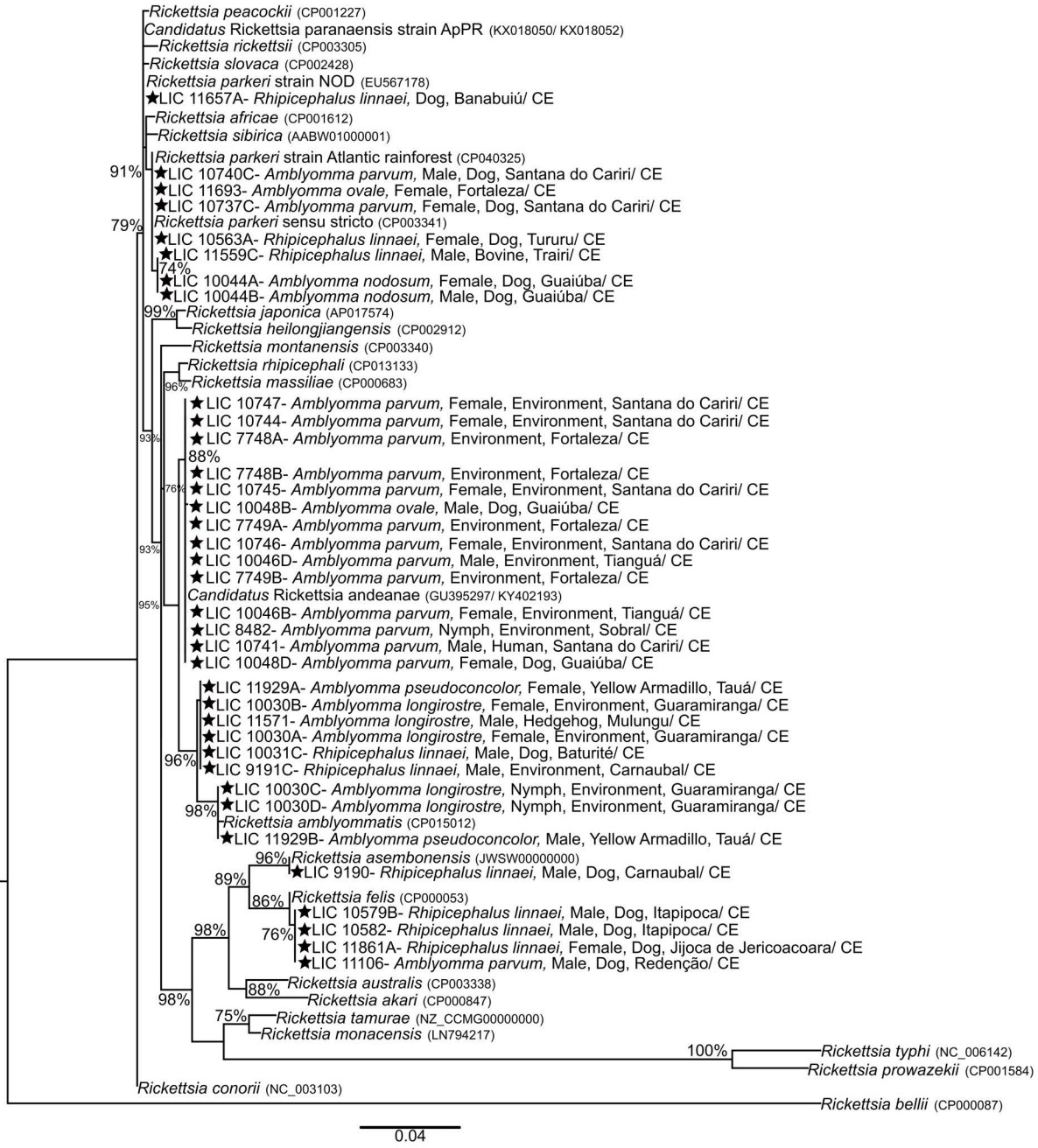

**Fig 1. Concatenated phylogeny, inferred through maximum likelihood analysis with the T92+G evolutionary model, of partial DNA sequences from the ompB + htrA genes of rickettsiae (411bp + 434bp) detected in ticks from Ceará, Brazil (highlighted with a star), in comparison to Rickettsia sequences retrieved from GenBank.** The numbers on the branches represent statistical support values (70% cut-off). Stars indicate sequences obtained in the present study.

Consequently, while it was possible to use the available data for the conserved *htrA* gene [31], the results may not fully reflect the detection of *Rickettsia parkeri* strain NOD in samples that only amplified the *ompB* fragment. Thus, considering the epidemiological knowledge of Spotted Fever in the studied region and the results from phylogenetic

**Table 3. Distribution of sequenced samples correlating vectors, hosts, and species of *Rickettsia spp*. identified during the period from 2017 to 2023.**

| Vectors | Hosts | *Rickettsia spp.* |
|---|---|---|
| *Rhipicephalus linnaei* | *Canis familiaris*<br>*Bos taurus* | *Rickettsia parkeri*<br>*Rickettsia amblyommatis*<br>*Rickettsia felis*<br>*Rickettsia asembonensis* |
| *Amblyomma parvum* | *Canis familiaris*<br>*Homo sapiens* | *Rickettsia parkeri*<br>*Rickettsia felis*<br>Ca. R. Andeanae |
| *Amblyomma longirostre* | *Coendou prehensilis* | *Rickettsia amblyommatis* |
| *Amblyomma nodosum* | *Canis familiaris* | *Rickettsia parkeri* |
| *Amblyomma ovale* | *Canis familiaris* | *Rickettsia parkeri* |
| *Amblyomma pseudoconcolor* | *Euphractus sexcinctus* | *Rickettsia amblyommatis* |

**Table 4. Distribution of *Rickettsia* Species, Associated Vectors, Hosts, and Localities in Ceará, Brazil (2017–2023).**

| Rickettsiae | Vectors | Hosts | Municipalities (n) |
|---|---|---|---|
| *Candidatus Rickettsia andeanae* | *A. parvum* | Environment | Santana do Cariri (n=4)<br>Fortaleza (n=4)<br>Tianguá (n=2)<br>Sobral (n=1) |
| | | *Canis familiaris* | Guaiuba (n=1) |
| | | *Homo Sapiens* | Santana do Cariri (n=1) |
| | *Rh. linnaei* | Environment | Carnaubal (n=1) |
| | | *Canis familiaris* | Baturité (n=1) |
| | *A. ovale* | *Canis familiaris* | Guaiuba (n=1) |
| *Rickettsia amblyommatis* | *A. longirostre* | Environment | Guaramiranga (n=4) |
| | | *Coendou prehensilis* | Mulungu (n=1) |
| | *A. pseudoconcolor* | *Euphractus sexcinctus* | Tauá (n=2) |
| | *Rh. linnaei* | *Canis familiaris* | Baturité (n=1) |
| *Rickettsia parkeri* | *A. nodosum* | *Canis familiaris* | Guaiuba (n=2) |
| | *A. parvum* | *Canis familiaris* | Santana do Cariri (n=2) |
| | *A. ovale* | - | Fortaleza (n=1) |
| | *Rh. linnaei* | *Canis familiaris* | Trairi (n=1) |
| | | *Bos taurus* | Tururu (n=1) |
| *Rickettsia felis* | *A. parvum* | *Canis familiaris* | Itapipoca (n=1) |
| | *Rh. linnaei* | *Canis familiaris* | Redenção (n=2)<br>Jijoca de Jericoacoara (n=1) |
| *Rickettsia asembonensis* | *Rh. linnaei* | *Canis familiaris* | Carnaubal (n=1) |

A.: Amblyomma, Rh.: Rhipicephalus.

analysis, a short but perceptible genetic distance was observed between samples LIC 10044B, LIC 10044A, and LIC 11559C and the *R. parkeri* Atlantic Rainforest strain samples. This proximity raises the hypothesis that these samples may be infected with *R. parkeri* strain NOD, rather than the Atlantic Rainforest strain indicated by the phylogenetic tree.

PLOS Neglected Tropical Diseases

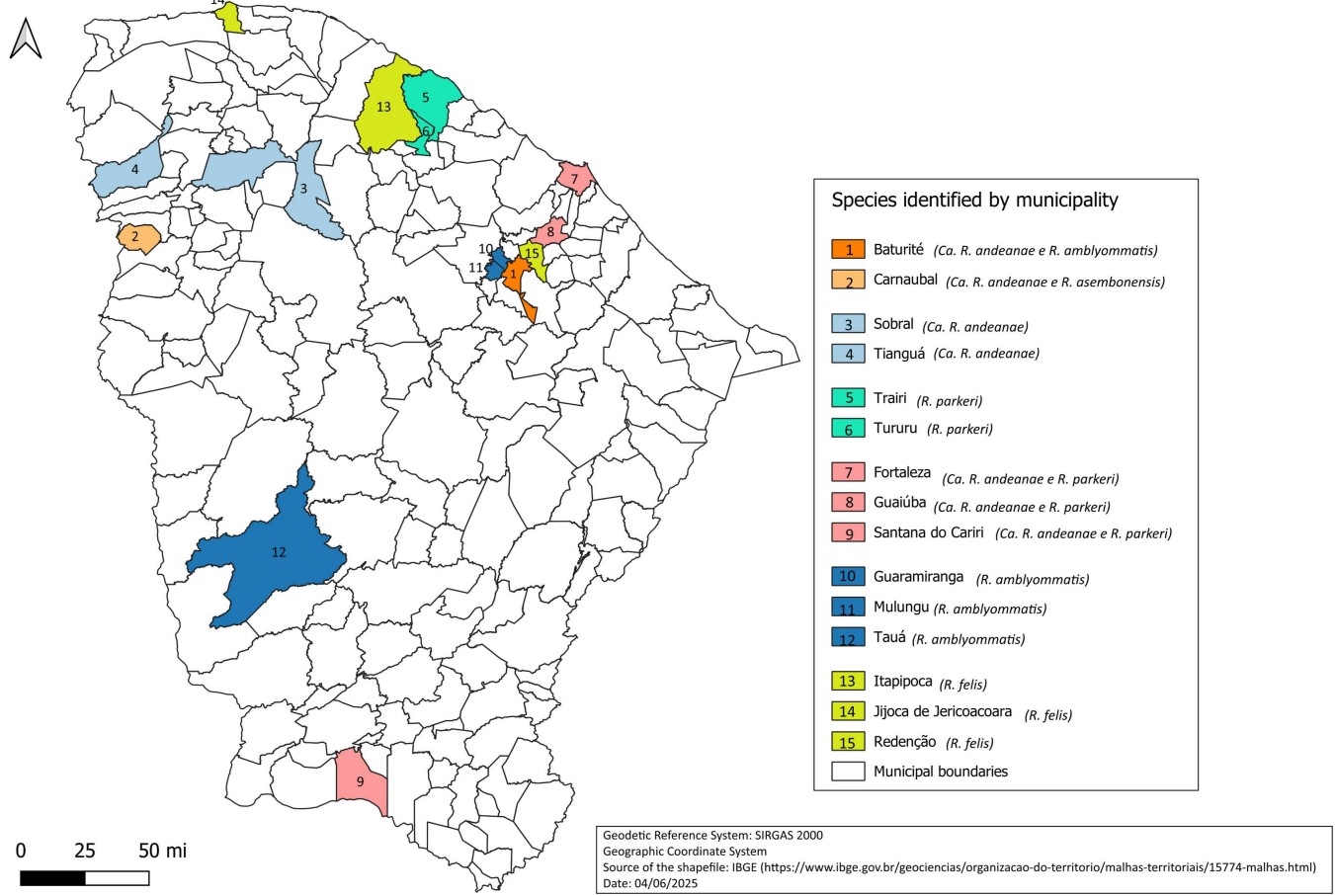

**Fig 2. Geographic Distribution of Rickettsia Species Identified in Tick Samples from Municipalities of Ceará, Brazil during the period from 2017 to 2023.** The map was created using QGIS 3.36 with geospatial layers publicly available from the Brazilian Institute of Geography and Statistics (IBGE). The coordinate reference system used was SIRGAS 2000/ UTM zone 24S. Scale bar and municipal boundaries are accurate to the dataset's resolution.https://www.ibge.gov.br/geociencias/organizacao-do-territorio/malhas-territoriais/15774-malhas.html.

Upon analyzing the results, it became evident that the tick species *Rhipicephalus linnaei* was the most frequently collected during the study (Table 2), with a total of 674 samples processed. From these, 45 (6.67%) successfully amplified during the screening phase conducted throughout the study period. However, due to the exhaustion of certain materials and the degradation of stored DNA, only 10 samples (1.48% of the total) were successfully sequenced. Notably, this species exhibited the greatest diversity of associated *Rickettsia* spp., including *Rickettsia parkeri* (Atlantic Rainforest strain), which, alongside *R. rickettsii*, is recognized as one of the primary causative agents of Spotted Fever in Brazil [2].

Although the finding of *R. parkeri* in *Rhipicephalus linnaei* does not necessarily serve as conclusive evidence of its potential as a vector for Spotted Fever, previous studies conducted in Brazil have demonstrated the ability of *Rickettsia* from the Spotted Fever Group (e.g., *Rickettsia rickettsii*) to be transmitted to vertebrate mammals through *Rh. linnaei* bites, as well as through transstadial and transovarial transmission. These studies have classified *Rhipicephalus linnaei* as a competent vector for Spotted Fever, at least in laboratory settings [9,32,33].

The findings of Piranda et al. [32] in a laboratory experiment demonstrated the ability of dogs to act as amplifiers of rickettsiae. However, it was shown that less than 40% of the ticks used in the experiment became infected when feeding on dogs infected with *Rickettsia rickettsii*. In parallel, the study by Pacheco et al. [9] found a 100% infection rate in

the offspring of naturally infected female *Rh. linnaei* collected directly from free-roaming dogs, demonstrating the natural maintenance capability of the rickettsiae's GFM enzootic cycle. Similarly, studies conducted in Mexico support these findings, with *Rh. sanguineus* s.l. (i.e., *Rh. linnaei*) considered the primary species associated with the transmission of Spotted Fever by *R. rickettsii* in the northern part of the country, with dogs as the main host [33,34].

Simultaneously, we also observed the presence of *Rickettsia parkeri* in *Amblyomma nodosum* and *Amblyomma ovale*, in agreement with the findings of Moerbeck et al. [12,28] in the state of Ceará. However, the presence of this agent in *Amblyomma parvum* had not yet been documented, making this the first report of an interaction between these species. This interaction is particularly noteworthy due to the low parasitic specificity and markedly anthropophilic behavior exhibited by this ixodid, as highlighted by Guglielmone and Robbins [35]. Nonetheless, caution is warranted when discussing this topic, given that the species is not currently recognized as a competent vector for *R. parkeri*, and the detection may reflect co-feeding transmission rather than true vector competence. Furthermore, it is also relevant that the infected *A. parvum* specimens were collected from hosts in the municipality of Santana do Cariri, where, to date, no species-specific confirmation of any *Rickettsia* has been reported [13].

These data, coupled with the relatively low number of *Amblyomma ovale* ticks collected and processed during the study (Table 2), the low prevalence of *Rickettsia parkeri* among these specimens—only one tick tested positive for the agent (3.22% of the total processed)— provide important context. Considering that the positive specimen was identified in a municipality where *R. parkeri* presence had already been previously confirmed (Fortaleza), suggest not only a potential role for *A. parvum* in the *R. parkeri* cycle but also a likely significant involvement of *Rhipicephalus linnaei* in the maintenance of the enzootic and epidemiological cycle of *R. parkeri*, at least in the state of Ceará. However, further studies are necessary to clarify and confirm this hypothesis.

**Candidatus Rickettsia andeanae.** First reported in Brazil in 2014 by Nieri-Bastos et al., *Ca. Rickettsia andeanae* is a bacterium from the spotted fever group (SFG) with a wide geographical distribution, found infecting different tick species in several countries such as Peru, Paraguay, Argentina, Chile, the United States, and Brazil [36–38]. In Brazil, this rickettsia is commonly associated with *Amblyomma parvum* ticks, being recorded in at least four Brazilian biomes: Cerrado, Pantanal, Atlantic Forest, and Caatinga [12,39,40], a finding corroborated by the present study, in which 13 (34.21%) of the 38 specimens of *A. parvum* collected were sequenced for the agent in question. Furthermore, its presence was also detected in two *Rhipicephalus linnaei* ticks and one *Amblyomma ovale* specimen, consistent with the findings of Krawczack et al. [38].

Although considered to have undetermined pathogenicity, Krawczak et al. [38] suggests that *Ca. R. andeanae* may not be pathogenic at all. This hypothesis is supported by the studies of Ferrari et al. [41], which demonstrated the bacterium's apparent inability to multiply in mammalian cells in laboratory experiments with Vero cells, and Grasperge et al. [42], which observed its apparent lack of transmission and reproduction in the skin of mice. Nonetheless, *Ca. R. andeanae* presents particularities that warrant further investigation, such as its association with various tick species and a potential negative correlation with *Rickettsia parkeri*, as suggested by Paddock et al. [43] and Krawczak et al. [38].

This correlation gains additional significance when considering that ixodid ticks are known to be unable to simultaneously transmit multiple *Rickettsia* species via vertical transmission. This has been demonstrated, for instance, by the exclusion or inhibition of the transovarial transmission of *Rickettsia rickettsii* by *Rickettsia peacockii* in *Dermacentor andersoni* [44], of *Rickettsia rhipicephali* by *R. montanensis* in *Dermacentor variabilis* [45] and of *R. rickettsii* by *Rickettsia bellii* in *Amblyomma dubitatum* [46]. Krawczak et al. [38] drew a similar inference when comparing the populations of *Rickettsia parkeri* and *Candidatus Rickettsia andeanae* in *A. tigrinum* in southern Brazil.

This phenomenon, called "rickettsial interference," occurs when the initial infection by one species of Rickettsia prevents or partially inhibits ovarian infection by another species. Although the exact mechanism of this interference is still not known [45], such microbiological interaction may play a significant role in the distribution of pathogenic rickettsial species, and the diseases associated with ixodid ticks.

Thus, due to the nature of the vectors associated with *Ca. R. andeanae*, especially its interaction with *A. parvum*, a vector of low specificity and considerable anthropophily, new studies are needed to confirm the organism's lack of pathogenicity, as well as its role as a possible inhibitor of pathogenic species.

**Rickettsia Amblyommatis.** The last member of the SFGR identified in the study, *Rickettsia amblyommatis* (known as *Candidatus Rickettsia amblyommii* until 2016), is a cosmopolitan rickettsia, transmitted both transovarially and transstadially, and can be found in a wide array of ticks throughout the Americas, including several species from the genera *Amblyomma*, *Rhipicephalus*, and *Dermacentor* [47,48].

Originally considered non-pathogenic due to the absence of clinical signs after inoculation in guinea pigs, it is now understood that its pathogenicity seems to be related to the infecting strain. More recent studies, which used different strains of the species, demonstrated clinical signs such as testicular enlargement, dermatitis, moderate fever, and behavioral changes, along with findings such as hepatic necrosis and increased antibody titers, indicating an immune response [48,49].

Regarding its ability to infect humans, Richardson et al. [49], in their review, highlight reports of immune reactions such as skin rash, erythema migrans, and fever following bites from *Amblyomma americanum* - the primary vector of *R. amblyommatis* in the United States - as well as epidemiological and diagnostic findings suggesting an association between these symptoms and infection by the agent.

This scenario supports the findings of Apperson et al. [50], who detected elevated IgG titers for *R. amblyommatis* in patients presenting with a disease similar to Spotted Fever, yet without reactivity to *R. rickettsii*. Nonetheless, both Apperson et al. [50] and Richardson et al. [49] contend that these observations do not definitively establish the pathogenicity of *R. amblyommatis* in humans and emphasize the need for targeted studies to clarify this, particularly given the frequent co-occurrence of different rickettsial species in the same regions

Furthermore, as appears to be the case with *Ca. R. andeanae* and other previously mentioned species, some studies indicate that *R. amblyommatis* may reduce or inhibit the persistence of infection and transmission of other rickettsiae in vectors (rickettsial interference), particularly *R. rickettsii* and *R. parkeri*. This could potentially reduce the intensity of infection in mammals, acting as a form of protective factor, although the existing studies on this matter remain inconclusive [49,51].

In light of these considerations, the detection of *Rickettsia amblyommatis* for the first time in the state of Ceará warrants attention, particularly due to its association with three different species of ixodid ticks: *Rh. linnaei*, *A. parvum*, and *A. pseudoconcolor*. Thus, the possible pathogenic potential of *R. amblyommatis*, the significant number of vectors already associated with it, and its possible inhibitory effect on other rickettsiae [49], underscore the need for further studies. Such investigations are essential to clarify the role of this bacterium as a potential etiological agent and its relevance in maintaining the biological cycles of other rickettsial species.

## Transitional group Rickettsiae

***Rickettsia felis*.** Originally described in 1991 and commonly associated with *Ctenocephalides felis* fleas, its primary vector and reservoir, *Rickettsia felis* is an organism with recognized pathogenicity, considered a common cause of febrile conditions in Africa, similar to murine typhus [52,53]. Due to the worldwide distribution of *C. felis*, its primary vector, and the detection of the agent in various species of ticks and mites across different parts of the globe, *R. felis* is considered a ubiquitous pathogen [54]. However, despite being confirmed as pathogenic and widely distributed, the biological characteristics and eco-epidemiology associated with the species are still not fully understood [55].

It is noteworthy that, despite its confirmed pathogenicity, broad geographic distribution, and association with various vectors, only a single case of *Rickettsia felis* has been documented in Brazil to date [56]. This is likely due to underreporting of cases, particularly considering that the clinical presentation caused by the agent is generally mild and nonspecific.

Such characteristics may lead both infected patients and healthcare professionals to deem further investigation of the underlying causes unnecessary. Thus, the need for more research on the pathogen's biology and its role in public health is evident, especially when considering the worldwide distribution of its vectors.

**Rickettsia asembonensis.** *Rickettsia asembonensis* is a bacterium that belongs to the group of *R. felis*-like organisms (RFLOs) within the Transitional Group of Rickettsia, transmitted by ticks and fleas [57]. Originally identified in Kenya in 2013, the species has since been detected in various arthropods collected from vertebrate hosts around the world, as noted by Loyola et al. [58].

Although understanding of its pathogenicity is still in its early stages, with infections caused by the organism generally considered asymptomatic, some studies have already pointed to a correlation between the detection of the agent in humans and cases of acute fever, suggesting a pathogenic potential, even in the Amazon Basin [58,59]. Furthermore, studies by Silva et al. [60] and Bitencourth et al. [61] have already linked its presence to ticks in Brazil, and the present study further underscores the need for attention to this bacterium, having found the rickettsia in association with *Rh. linnaei* ticks on Ceará state for the first time, thus showing its expansion.

## Conclusion

*Rhipicephalus linnaei, Amblyomma parvum*, *Amblyomma longirostre*, *Amblyomma nodosum*, *Amblyomma ovale*, and *Amblyomma pseudoconcolor* are infected ectoparasite species found in the studied region.

This study represents the first report of the circulation of *Rickettsia amblyommatis* and *Rickettsia asembonensis* in potential vectors in the state of Ceará.

It also marks the first documented association between *Amblyomma parvum* and *Rickettsia parkeri*.

The presence of *Candidatus Rickettsia andeanae* and *Rickettsia amblyommatis* may be linked to a lower frequency of vectors infected with *R. rickettsii* and *R. parkeri* in the region. While this possible interaction among *Rickettsia* species remains speculative, it warrants further investigation, as understanding such dynamics could be important for refining risk assessments of spotted fever group rickettsioses and informing public health surveillance strategies.

Although the pathogenicity of *R. amblyommatis* remains officially undetermined, studies suggest its potential pathogenicity, highlighting the need for further evaluation of its role as a possible etiological agent.

## Acknowledgments

We would like to acknowledge Dr. Flávio Fernando Batista Moutinho (Universidade Federal Fluminense - Departamento de Saúde Coletiva Veterinária e Saúde Pública) for his valuable contribution preparing the geographic visualization presented in Fig 2, and the Veterinary Student Clara Maciel (Universidade Federal Fluminense) for helping with the preparation and processing of the samples utilized during this study.

## Author contributions

**Conceptualization:** João F. Audi Gazeta, Ana Beatris Pais Borsoi, Karla Bitencourth, Mariana Guimarães Côrtes, Ingrid Benevides Machado, Robson Cavalcante, Gilberto Salles Gazeta, Nathalie Costa da Cunha.

**Formal analysis:** João F. Audi Gazeta, Beatriz Pinheiro Melo da Silva.

**Investigation:** Beatriz Pinheiro Melo da Silva.

**Methodology:** João F. Audi Gazeta, Ana Beatris Pais Borsoi, Karla Bitencourth, Mariana Guimarães Côrtes, Ingrid Benevides Machado, Beatriz Pinheiro Melo da Silva, Robson Cavalcante, Nathalie Costa da Cunha.

**Project administration:** João F. Audi Gazeta.

**Supervision:** Ana Beatris Pais Borsoi, Gilberto Salles Gazeta, Nathalie Costa da Cunha.

**Visualization:** Ana Beatris Pais Borsoi, Karla Bitencourth, Mariana Guimarães Côrtes, Ingrid Benevides Machado, Robson Cavalcante, Gilberto Salles Gazeta, Nathalie Costa da Cunha.

**Writing – original draft:** João F. Audi Gazeta.

**Writing – review & editing:** João F. Audi Gazeta, Ana Beatris Pais Borsoi, Karla Bitencourth, Nathalie Costa da Cunha.

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
