## [Decision Letter · Decision Letter 0]

1 May 2025

The Mosaic of Rickettisae in Caatinga biome, Brazil: New interactions between Rickettsia parkeri and Amblyomma Parvum

Dear Dr. Audi Gazeta,

Thank you for submitting your manuscript to PLOS Neglected Tropical Diseases. After careful consideration, we feel that it has merit but does not fully meet PLOS Neglected Tropical Diseases's publication criteria as it currently stands. Therefore, we invite you to submit a revised version of the manuscript that addresses the points raised during the review process.

Please submit your revised manuscript within 60 days Jun 30 2025 11:59PM. If you will need more time than this to complete your revisions, please reply to this message or contact the journal office at plosntds@plos.org. Please include the following items when submitting your revised manuscript:

We look forward to receiving your revised manuscript.

Kind regards,

Yazid Abdad

Guest Editor

Stuart Blacksell

Section Editor

Shaden Kamhawi

co-Editor-in-Chief

Paul Brindley

co-Editor-in-Chief

**Journal Requirements:**

At this stage, the following Authors/Authors require contributions: João Felipe Audi Gazeta. Please ensure that the full contributions of each author are acknowledged in the "Add/Edit/Remove Authors" section of our submission form.

- ® on page: 8.

4) We note that your Data Availability Statement is currently as follows: "The data supporting these finds will be in the process of deposition on GenBank within the week. Any further doubts and inquiries, my e-mail is available (joaofgazeta@hotmail.com)". Please confirm at this time whether or not your submission contains all raw data required to replicate the results of your study. Authors must share the “minimal data set” for their submission. PLOS defines the minimal data set to consist of the data required to replicate all study findings reported in the article, as well as related metadata and methods (https://journals.plos.org/plosone/s/data-availability#loc-minimal-data-set-definition).

**Reviewers' Comments:**

Reviewer's Responses to Questions

**Key Review Criteria Required for Acceptance?**

**Methods**

-Are the objectives of the study clearly articulated with a clear testable hypothesis stated?

-Is the study design appropriate to address the stated objectives?

-Is the population clearly described and appropriate for the hypothesis being tested?

-Is the sample size sufficient to ensure adequate power to address the hypothesis being tested?

-Were correct statistical analysis used to support conclusions?

-Are there concerns about ethical or regulatory requirements being met?

Reviewer #1: From the description, the samples appear to be from archived collections, but a few clarifications are needed:

1. Please provide a map indicating where the samples were collected and where the laboratory work was conducted. The locations seem to be quite far apart, so additional clarification is required to confirm the validity of identification process.

2. Are the same researchers involved in both the analysis and the sample collection? How can you verify that the methods used were flannel dragging and direct picking?

3. How were the tick samples preserved after collection?

4. When was the taxonomic identification performed? Was the identification conducted after the sample collection between 2017 and 2023, or was it done recently after the samples were retrieved from the repository?

5. How were the samples stored, and at what temperature?

6. How the samples were transported from the repository to the lab?

7. Some of the paragraphs lack clarity. Please rewrite it to clarify the details especially the new PCRs method.

Detailed comments are provided in the attachment.

Reviewer #2: The study objective is poorly formulated.

The study protocol is suitable for the molecular identification of Rickettsia species from ticks in Caatinga, Brazil.

The sample sizes are sufficient to verify Rickettsia in this study area.

Reviewer #3: The use of PCR assays is appropriate but the rationale for how and why they were used is not clearly explained. The section that starts: “The shift in primers used was due to their characteristics…” is confusing, because the second set of primers used for sequencing are the nested ones, which would be more sensitive, based on the description, than the assays used for screening. The statement “Thus, new PCRs for both and new positive samples were conducted using nested primers, chosen for their higher detection sensitivity…” At least with respect to the use of ompB, it differentiates between similar SFG species better than the gltA or htrA protocols, which I would think would be its primary benefit in using it for the sequencing reactions here.

**Results**

-Does the analysis presented match the analysis plan?

-Are the results clearly and completely presented?

-Are the figures (Tables, Images) of sufficient quality for clarity?

Reviewer #1: The results presented focus on the distribution and analysis of molecular detection of Rickettsia spp. infections in vectors. Information about the hosts was also included; however, there was no additional information regarding environmental factors. If you do not plan to discuss the variations and differences related to biomes, microregions, and mesoregions, please remove this sentence. Detailed comments can be found in the attachment.

Reviewer #2: The results presented in the study are unclear and require restructuring for ease of reading.

Fig. 1 is unclear. Please tell us what new information this figure adds to the study.

Reviewer #3: Results

1.Table 1 and Table 2 are redundant—Table 1 is just the row tabulations of Table 2. I believe Table 1 can be eliminated.

2.Recommend using common host names along with scientific for host names in the tables.

3.Table 3: Headings in Portuguese

4.In the statement “Of these 37 sequenced samples, 24 were analyzed for the ompB gene, from a total of 25 amplified by Nested PCR, and 21 were analyzed for the htrA gene, from a total of 23 amplified by Nested PCR.”: I am confused about what this means. Does this mean that 25 of the ompB samples produced bands and 24 were sequenced successfully, and that 23 of the htrA produced bands, of which 21 were sequenced successfully, or something else? Please clarify. It would also be helpful to indicate the number of samples that were sequenced for both ompB and htrA.

5.“Notably, of the 37 sequences obtained…”: This should be “samples that were successfully sequenced”, not number of sequences, correct?

**Conclusions**

-Are the conclusions supported by the data presented?

-Are the limitations of analysis clearly described?

-Do the authors discuss how these data can be helpful to advance our understanding of the topic under study?

-Is public health relevance addressed?

Reviewer #1: The overall discussion and conclusion was satisfactory, focusing on the disease distributions found in this study and their comparisons to other studies. However, this study solely focused on detecting Rickettsia infections in archived tick samples. A revised title should accurately represent the scope of the study.

Reviewer #2: The results could be of interest for public health.

Reviewer #3: 1.The discussion is very comprehensive, but the depth that each individual pathogen is described exceeds what is necessary for this manuscript. For example, it is absolutely worth discussing the importance Ca. R. andeanae in the transmission of other rickettsial organisms, but the description of rickettsial interference and breadth of references used to discuss it here is more than warranted.

2.Missing from the discussion is a correlation between the species identified here and the epidemiology of rickettsial disease in the region. I also feel that at least a short section putting the results in context of disease in this region compared to the rest of Brazil would be useful to the reader.

3.“Although the finding of R. parkeri in Rhipicephalus linnaei does not necessarily serve as conclusive evidence of its potential as a vector for Spotted Fever, previous studies conducted in Brazil have demonstrated the ability of Rickettsia from the Spotted Fever Group….”: The source of the tick is important here in the discussion of whether R. linnaei is a potential vector. If sourced from an animal, a positive result may be due to the presence of the bloodmeal, which warrants mention.

4.The paragraph starting “The findings of Piranda et al.” is of questionable relevance because R. rickettsii was not detected in this study

5.“…but also a likely significant involvement of Rhipicephalus linnaei in the maintenance of the enzootic and epidemiological cycle of R. parkeri, at least in the state of Ceará…” Please clarify why there is likely significant involvement of R. linnaei in R. parkeri transmission, as the data given here don’t seem to directly support that.

**Editorial and Data Presentation Modifications?**

Reviewer #1: This manuscript requires major revisions. The overall analyses conducted are adequate for publication; however, the authors need to restructure the entire manuscript from the title to the conclusion. The title somewhat overclaims the data that are not presented in the manuscript.

Reviewer #2: (No Response)

Reviewer #3: 1.Figure 1 is hard to read, both because of lines not being spaced well and because it’s really hard to find the most closely correlated reference sequence in the text. It would help to bold or otherwise make very obvious the reference sequences so that the identity of the study sequences can be better understood.

2.A map of the location of the state and maybe municipalities within the state would be very helpful.

**Summary and General Comments**

Reviewer #1: The authors presented their findings on the prevalence of Rickettsia spp. infections from tick samples obtained from various hosts. This information could be valuable to local stakeholders. However, the presentation of the results lacks clarity. Out of a total of 3,098 ticks collected, only 1,359 were processed, and all positive samples were sourced from localities in Ceará. Additionally, there was no analysis regarding environmental factors, and the interaction between Rickettsia parkeri and Amblyomma parvum was only briefly mentioned in one paragraph. The authors should consider revising the title for better clarity. Detailed comments can be found in the attachment.

Reviewer #2: The manuscript “The Mosaic of Rickettisae in Caatinga Biome, Brazil: New Interactions between Rickettsia parkeri and Amblyomma Parvum” is an interesting research work and an original idea, which provides information on Rickettsia species from ticks in Caatinga, Brazil. The manuscript requires extensive revision.

Reviewer #3: This manuscript describes the detection of a number of rickettsial species in ticks in the Northeast of Brazil, via testing of a relatively large number of ticks collected through follow-up investigations of human rickettsiosis cases. The methods are appropriate although I have some concerns about how they were explained and justified, and recommend clarification. Overall, this is valuable information but would be more useful to the reader with additional context.

PLOS authors have the option to publish the peer review history of their article (what does this mean? ). If published, this will include your full peer review and any attached files.

**Do you want your identity to be public for this peer review?** For information about this choice, including consent withdrawal, please see our Privacy Policy .

Reviewer #1: No

Reviewer #2: No

Reviewer #3: No

**Figure resubmission:**

**Reproducibility:**



---

## [Decision Letter · Decision Letter 1]

9 Sep 2025

Response to Reviewers
Revised Manuscript with Track Changes
Manuscript

Shaden Kamhawi

co-Editor-in-Chief

Paul Brindley

co-Editor-in-Chief

**Additional Editor Comments:**
**Reviewers' comments:**

**Key Review Criteria Required for Acceptance?**

**Methods:**

-Are the objectives of the study clearly articulated with a clear testable hypothesis stated?

-Is the study design appropriate to address the stated objectives?

-Is the population clearly described and appropriate for the hypothesis being tested?

-Is the sample size sufficient to ensure adequate power to address the hypothesis being tested?

-Were correct statistical analysis used to support conclusions?

-Are there concerns about ethical or regulatory requirements being met?

Reviewer #1: The method has undergone significant improvement, now following a clear and easy-to-understand flow.

Reviewer #4: The overall methods have been improved and are clear. One potential important missing element is information on IACUC or any veterinary health regulatory board approval. This may not be required in this instance, but I know some organizations are more stringent than others. If a veterinary health board approval was obtained, please add it the first part of the methods section.

Reviewer #5: - Need more clarity and details

- Needs to be re-written with clear subheadings on setting, sampling procedures (collection, storage, transport etc) , research design, sample processing (clearly differentiating morphological identification, physical processing of ticks, primary molecular detection, DNA sequencing etc)

- There is no mention of sequence submission to GeneBank and acquisition of accession number?

- - There is no mention of anything on ethics and approvals, or any sort of waivers if available. This needs to be mentioned.

**Results:**

-Does the analysis presented match the analysis plan?

-Are the results clearly and completely presented?

-Are the figures (Tables, Images) of sufficient quality for clarity?

Reviewer #1: The results are well presented, organized, and easier to understand. I just need a few more clarifications as below:

How is the confirmation of the host species conducted?

Are ectoparasites identified solely through morphological identification?

Reviewer #4: This section has been much improved and no further recommendations are noted.

Reviewer #5: - Elaborate and adequate

- Figure of Phylogenetic tree need a better and more clear picture

**Conclusions:**

-Are the conclusions supported by the data presented?

-Are the limitations of analysis clearly described?

-Do the authors discuss how these data can be helpful to advance our understanding of the topic under study?

-Is public health relevance addressed?

Reviewer #1: Why did u suggested this conclusion?

"The presence of Candidatus Rickettsia andeanae and Rickettsia amblyommatis may be linked to a lower frequency of vectors infected with R. rickettsii and R. parkeri in the region".

I recommend not including this sentence in the conclusion, as the study focuses solely on reporting the prevalence of the pathogens.

Reviewer #4: Two important notes/comments are missing from the discussion section. The authors need to specifically discuss the limitation that they had poor genetic material in older ticks (as mentioned in the results section) and that this could lead to an underreporting or missed cases. Additionally, the authors should discuss the lack of R. rickettsii in their ticks. I was surprised they didn't find this pathologically important species, and a brief discussion should be noted (pontificate why this species may not be in the local environment).

Reviewer #5: (No Response)

**Editorial and Data Presentation Modifications?**

Reviewer #1: Minor revision.

Reviewer #4: Based on the author's response to initial reviewers, the article has been greatly improved.

There are a few places where minor grammatical edits are needed (e.g. missing "Rickettsia" in front of 'spp' in the abstract, not italicizing scientific names at a few places in the manuscript, or capitalizing species names [e.g. Ca. R. Andeanae in table 3]. These are minor and likely an artifact of the tracked changes to accept changes process.

Another minor comment. The authors go back and forth between "Spotted Fever" and "SF". I would recommend they use the acronym "SFGR" to be consistent with the scientific literature (for spotted fever group Rickettsia). Please consistently use the acronym in subsequent text once it is introduced (e.g. if introduced in the introduction section, please continue to use in the methods, results and discussion).

Reviewer #5: (No Response)

**Summary and General Comments:**

Reviewer #1: The manuscript has improved significantly based on reviewer feedback. However, a few clarifications are still needed. Please refer to the specific comments in the attachment and address them accordingly.

Reviewer #4: None.

Reviewer #5: The authors need to run through and read the manuscript in view of checking all spellings, use of capital letters for words in between sentences since a few of these were observed. The manuscript also need to be formatted as per the author guideline. Even some track changes are still visible.

PLOS authors have the option to publish the peer review history of their article (what does this mean? ). If published, this will include your full peer review and any attached files.

**Do you want your identity to be public for this peer review?** For information about this choice, including consent withdrawal, please see our Privacy Policy .

Reviewer #1: No

Reviewer #4: No

Reviewer #5: **Yes: ** Tshokey Tshokey

**Figure resubmission:**

**Reproducibility:** To enhance the reproducibility of your results, we recommend that authors of applicable studies deposit laboratory protocols in protocols.io, where a protocol can be assigned its own identifier (DOI) such that it can be cited independently in the future. Additionally, PLOS ONE offers an option to publish peer-reviewed clinical study protocols. Read more information on sharing protocols at https://plos.org/protocols?utm_medium=editorial-email&utm_source=authorletters&utm_campaign=protocols

---

## [Editor Report · Decision Letter 2]

21 Oct 2025

Dear MSC Audi Gazeta,

We are pleased to inform you that your manuscript 'The Mosaic of Rickettisae from the Caatinga Biome, Brazil: New Records and First Interactions' has been provisionally accepted for publication in PLOS Neglected Tropical Diseases.

Best regards,

Yazid Abdad

Guest Editor

Stuart Blacksell

Section Editor

Shaden Kamhawi

co-Editor-in-Chief

Paul Brindley

co-Editor-in-Chief

---

## [Editor Report · Acceptance letter]

Dear MSC Audi Gazeta,

We are delighted to inform you that your manuscript, "The Mosaic of Rickettisae from the Caatinga Biome, Brazil: New Records and First Interactions," has been formally accepted for publication in PLOS Neglected Tropical Diseases.

Best regards,

Shaden Kamhawi

co-Editor-in-Chief

Paul Brindley

co-Editor-in-Chief
